# Reasons for reductions in routine childhood immunisation uptake during the COVID-19 pandemic in low- and middle-income countries: A systematic review

**Alexandra M. Cardoso Pinto**[1]*, **Sameed Shariq**[1], **Lasith Ranasinghe**[2], **Shyam Sundar Budhathoki**[3], **Helen Skirrow**[3], **Elizabeth Whittaker**[4], **James A. Seddon**[4,5]

**1** School of Medicine, Imperial College London, London, United Kingdom, **2** Academic Foundation Doctor, Imperial College London, London, United Kingdom, **3** Department of Primary Care and Public Health, School of Public Health, Imperial College London, London, United Kingdom, **4** Department of Infectious Disease, Imperial College London, London, United Kingdom, **5** Desmond Tutu TB Centre, Department of Paediatrics and Child Health, Stellenbosch University, Cape Town, South Africa

* alexandra.cardoso-pinto18@imperial.ac.uk

## Abstract

The coronavirus disease 2019 (COVID-19) pandemic has resulted in a substantial decline in routine immunisation coverage in children globally, especially in low- and middle-income countries (LMICs). This study summarises the reasons for disruptions to routine child immunisations in LMICs. A systematic review (PROSPERO CRD42021286386) was conducted following PRISMA 2020 guidelines. Six databases were searched: MEDLINE, Embase, Global Health, CINAHL, Scopus and MedRxiv, on 11/02/2022. Observational and qualitative studies published from January 2020 onwards were included if exploring reasons for missed immunisations during the COVID-19 pandemic in LMICs. Study appraisal used National Heart, Lung, and Blood Institute and Critical Appraisal Skills Programme tools. Reasons for disruption were defined with descriptive codes; cross-sectional (quantitative) data were summarised as mean percentages of responses weighted by study population, and qualitative data were summarised narratively. A total of thirteen studies were included describing reasons behind disruptions; 7 cross-sectional (quantitative), 5 qualitative and 1 mixed methods. Seventeen reasons for disruptions were identified. In quantitative studies (total respondents = 2,853), the most common reasons identified were fear of COVID-19 and consequential avoidance of health centres (41.2%, SD ±13.3%), followed by transport challenges preventing both families and healthcare professionals from reaching vaccination services (11.1% SD ±16.6%). Most reasons stemmed from reduced healthcare-seeking (83.4%), as opposed to healthcare-delivery issues (15.2%). Qualitative studies showed a more even balance of healthcare-seeking (49.5%) and healthcare-delivery issues (50.5%), with fear of COVID-19 remaining a major identified issue (total respondents = 92). The most common reasons for disruption were parental fear of COVID-19 and avoidance of health services. Health systems must therefore prioritise public health messaging to encourage vaccine uptake and recovery of missed immunisations.

**Data Availability Statement:** All relevant data are within the paper and its Supporting Information files.

**Funding:** This study was supported by Imperial College Open Access Fund in the form of open access publication funding. This study was also supported by UK Medical Research Council (MRC) and the UK Department for International Development (DFID) under the MRC/DFID Concordat agreement (MR/R007942/1) in the form of a Clinician Scientist Fellowship awarded to JS. The funders had no role in study design, data collection and analysis, decision to publish, or preparation of the manuscript.

**Competing interests:** Shyam Sundar Budhathoki is an Academic Editor for PLOS Global Public Health but was not involved in any editorial decisions for this manuscript. All other authors have declared that no competing interests exist.

## Background

The Coronavirus Disease 2019 (COVID-19) pandemic has changed the lives of children globally. Although children have largely been spared the consequences of severe clinical manifestations of COVID-19, they have been subject to several significant indirect effects of the pandemic [1–5]. This includes the impact of the pandemic on healthcare, and in particular immunisation services. Studies have reported substantial declines in administration of routine childhood immunisations since March 2020 [6, 7]. Such declines have resulted in significant setbacks in elimination of vaccine-preventable diseases such as measles and polio [8, 9]. Drops in routine vaccines also raise concern over increasing numbers of outbreaks, rises in health costs and increased child mortality [10].

Between the years 2000–2019, low- and middle-income countries (LMICs) achieved substantial increases in vaccination coverage. For example, the first dose of measles-containing vaccine (MCV-1) and third dose of the diphtheria-tetanus-pertussis (DTP-3) combined vaccine each increased by 14 percentage points during this time, both reaching 86% coverage globally [11]. Nevertheless, even prior to the COVID-19 pandemic, many children remained unvaccinated; in 2019, UNICEF estimates that 13 million children received no dose of the DTP vaccine–this value increased by 39% in 2021 [11].

Declines in immunisations have been found to be greatest in low or middle-income settings, compared to high-income settings [12]. This is also where the burden of vaccine-preventable diseases is highest and vaccine coverage lowest [2, 13]. This may, in part, be due to the reallocation of resources towards efforts to manage COVID-19, alongside measures to control its spread, which led to the closure of routine services, including immunisations [2, 14]. However, even in countries where vaccination programmes continued, disruptions were still reported [15, 16].

A previous systematic review summarised the extent of disruption to vaccination services in LMICs and found a decline of over 10% in all vaccines universally recommended by the World Health Organization (WHO) [7]. Data for this study ended in 2020. However, data from the WHO has suggested that declines persisted throughout 2021, with little evidence of recovery [17]. As nations shift focus towards recovery of immunisation coverage, it is essential to understand reasons behind non-attendance and non-delivery of vaccination, as this enables planning of effective catch-up programmes. Additionally, understanding factors contributing to this decline is fundamental to prevent similar future disruptions in pandemic-like contexts. Therefore, and following on from the findings of the previous systematic review [7], the aim of this study was to summarise the reasons behind disruptions to routine child immunisations in LMICs during the COVID-19 pandemic.

## Methods

A systematic review of published and pre-print literature was performed, following PRISMA 2020 guidelines (**S1 Checklist**) [18, 19].

### Ethics statement

As this work is a systematic review, ethics approval was not required.

### Search strategy and inclusion criteria

The search strategy is analogous to one described in a previous systematic review, summarising the magnitude of disruption to routine child vaccinations in LMICs [7].

**Table 1. Summary of inclusion and exclusion criteria for studies reporting levels of disruption and/or reasons for disruptions.**

| Inclusion | Exclusion |
|---|---|
| Qualitative or observational primary research studies | Non-primary research studies: reviews, editorials, meeting summaries |
| Data from LMICs defined following the latest World Bank income-level classification. May contain data from HICs, if this is reported separately to that of LMICs. | Studies combining LMIC and HIC data. |
| Data relating to childhood immunisations services. May contain data from other health services if this is reported separately to immunisation data. | Studies combining multiple health services, including adult immunisations. |
| Studies reporting quantitative and/or qualitative data on reasons behind disruptions to immunisation programmes | Reasons for vaccine hesitancy in general, including prior to the COVID-19 pandemic |
| Data must have been gathered during, and in relation to, the COVID-19 pandemic | |
| Language: English, Portuguese, Spanish, French* | |
| Published from 2020 onwards | |

LMIC: low- and middle-income country; HIC: high-income country

*Language restrictions were only applied at full-text stage. Studies not in any of the listed languages were translated to English using Google Translate.

Six databases were searched on the 11[th] of February 2022: Medline, EMBASE and Global Health via Ovid, CINAHL and Scopus, and MedRxiv using R code [20, 21]. Results were limited to studies published from January 2020 onwards. The search strategy was based on three domains: COVID-19, immunisation, and routine vaccines or vaccine-preventable diseases. Each domain contained relevant keywords, including variations, and subject headings (see **S1 Text** for full search strategies). The references of all relevant articles–including eligible studies, reviews, letters, editorials, commentaries, and conference abstracts–were also screened for inclusion.

Primary research studies exploring reasons for disruptions to immunisation programmes were included, where 'reasons for disruptions' are defined as any factor that may have contributed to declines in routine immunisations, as reported by studies (**Table 1**).

## Screening and selection

Search duplicates were removed using EndNote 20 and Covidence. Initial screening was performed by title alone, followed by abstract. Full texts were then reviewed for eligibility. Screening was performed by two reviewers independently and discrepancies resolved by consensus.

## Data extraction and study quality assessment

Data were extracted from included studies using a pre-defined Microsoft Excel spreadsheet, which included: publication details (doi, authors, title and year of publication), study design, method of data collection, time periods for data collection, location(s) of study, sample size, population, sampling methods, analysis methods and outcomes. Each outcome (i.e., each reason for disruption to routine immunisation) was extracted as reported by the study, along with the reporting stakeholder and its count, for cross-sectional studies. Extraction was performed by two reviewers independently, with discrepancies resolved by a third reviewer. Study quality was assessed using National Heart, Lung and Blood Institute (NHLBI) checklist for observational studies and Critical Appraisal Skills Programme (CASP) for qualitative studies [22, 23]. These tools were chosen as they were found sufficiently comprehensive to cover a range of

possible study designs at the time the protocol was designed. Assessments were performed by two reviewers independently, with discrepancies resolved by consensus. Reviewers assessed the quality of studies against the relevant checklist, noting if each criterion was met, not meet, or not applicable, followed by an overall quality score (poor, fair, good).

## Data analysis and synthesis

Following extraction, each reported reason was given a descriptive code, allowing similar causes from separate studies to be merged. Coding was performed by two reviewers independently and discussed until consensus was reached. For cross-sectional (quantitative) studies, the percentage of participants reporting each reason was calculated for each study and the mean (± standard deviation), weighted by sample population, was determined. Qualitative studies were not included in this analysis and were instead summarised narratively.

Coded reasons were further classified into health-seeking (factors affecting individuals' demand for immunisations) and health-delivery (health-services or healthcare professionals (HCPs) being unable to deliver immunisations) issues. For this analysis, in qualitative studies that did not report counts, every participant was assumed to have reported each reason, given the low numbers of participants and to ensure frequencies remained integers. Total responses were also subdivided by the reporting stakeholder and WHO World Region of the study.

## Registration

This systematic review was prospectively registered on PROSPERO (CRD42021286386). Changes to the protocol are described in **S2 Text**.

## Results

Following the screening of 7,690 studies, 13 were included (**Fig 1**).

A total of 13 studies [24–36] reporting reasons for disruptions to routine childhood immunisation were identified, all of which took place in 2020 (**Tables 2 and 3**). Studies employed cross-sectional and quantitative methods (n = 7), qualitative (n = 5) or mixed-methods (n = 1). Sources of data included questionnaires (n = 9), interviews (n = 5) and focus groups (n = 2), with participants including parents or caregivers (n = 9) and HCPs (n = 7).

Overall quality of studies was fair to good; several studies did not report participation rates, had unclear methods of synthesis and analysis, and lacked reflexivity statements (**Fig 2**).

### Cross-sectional (quantitative) studies

There were 3,160 total responses from 2,853 respondents across all studies, with 81.3% respondents being parents or caregivers and 18.7% being HCPs (**Fig 3**). WHO world region with most respondents was South-East Asia (49.9%). Approximately 83.4% of responses referred to healthcare-seeking challenges and 15.2% referred to healthcare-delivery challenges, the remaining being non-applicable. A total of 74 reasons were extracted and from these a total of 17 unique reasons for disruptions to routine immunisation were coded (see **S1 Table** for definitions of each code).

Overall, the most reported reason for missing immunisations was fear of COVID-19, including fear of contracting COVID-19 and fear of children contracting COVID-19, particularly in healthcare settings (**Fig 4**). Participants were also concerned about the prospect of having to self-isolate if they contracted, or were in contact with, COVID-19. The next most frequently reported reason was lockdown policies, including government advice to stay home

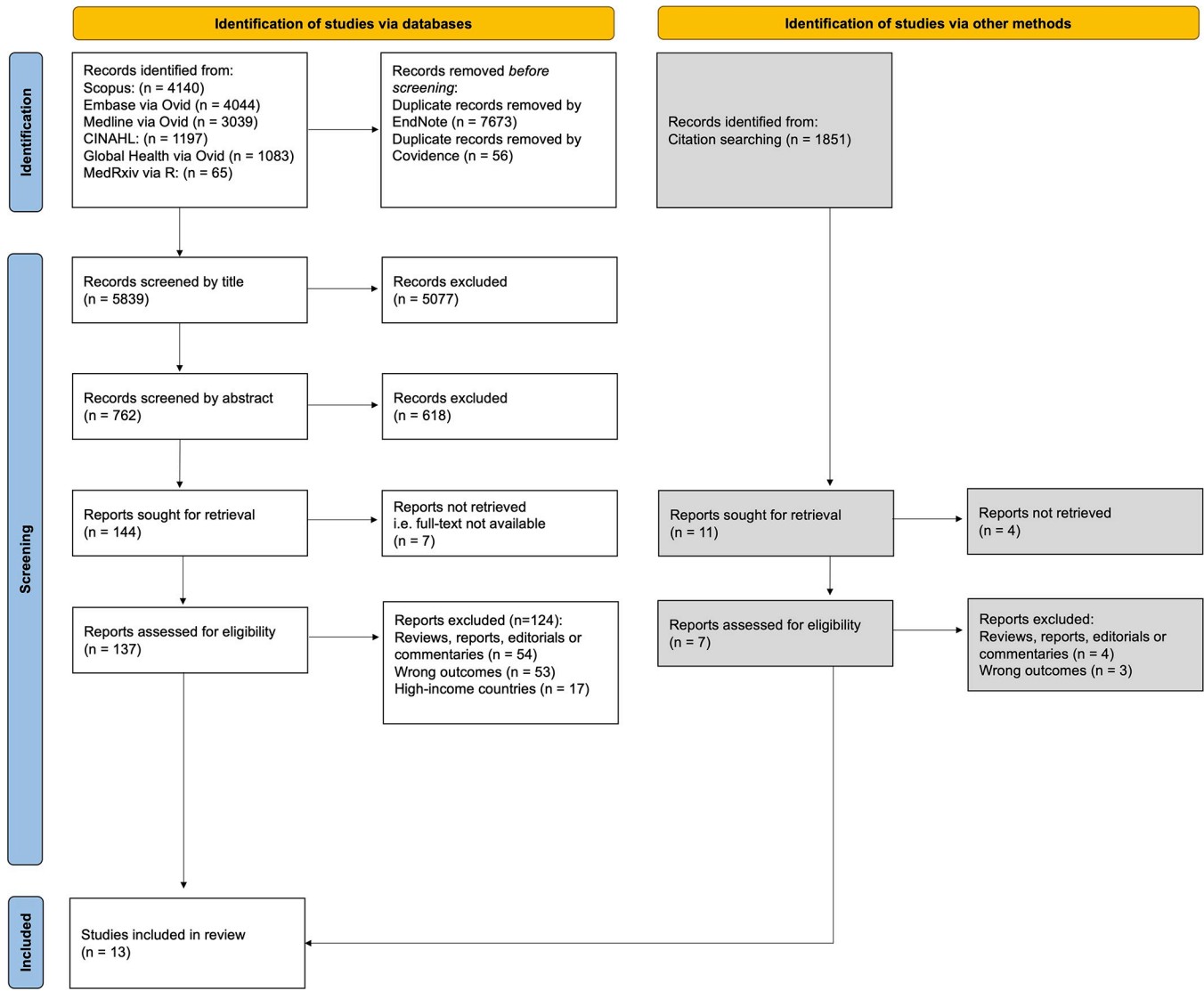

**Fig 1. PRISMA flowchart outlining identification and screening process for studies included in the review.**

and avoid public spaces. Unavailability of transport services, which limited the ability of both service-users and HCPs to reach immunisation services, was the third most reported reason for not attending immunisation. The most frequently reported health-delivery challenge was the closure of immunisation services, closely followed by challenges scheduling appointments and insufficient staffing, often due to increased staff demand for COVID-19-related services (**Fig 5**). While many coded causes were entirely classified as either health-seeking or health-delivery issues, some were classified with both. For instance, transport challenges which affected HCPs ability to attend work was classified health-delivery issue, whereas transport challenges that described patients being unable to attend services were classified as health-seeking issues. Similarly, fear of infection could apply to both HCPs and patients–although the vast majority, 93.6%, applied to parents–as could financial challenges, and scheduling challenges.

**Table 2. Summary of study characteristics for studies reporting reasons for disruptions to routine immunisations (n = 13).**

|  | Study characteristic* | Number of studies (%) | Study reference |
|---|---|---|---|
| **Study design** | Cross-sectional (quantitative) | 7 (53.8) | [25, 29–31, 33–35] |
|  | Mixed methods (quantitative and qualitative) | 1 (7.7) | [27] |
|  | Qualitative | 5 (38.5) | [24, 26, 28, 32, 36] |
| **WHO World region** | African Region (AFR) | 4 (30.8) | [24, 25, 28, 36] |
|  | Region of the Americas (AMR) | 2 (14.4) | [31, 36] |
|  | South-East Asian Region (SEAR) | 8 (61.5) | [26, 27, 30–32, 34–36] |
|  | Eastern Mediterranean Region (EMR) | 3 (23.1) | [29, 31, 33] |
|  | Western Pacific Region (WPR) | 1 (7.7) | [31] |
|  | European Region (EUR) | 0 | - |
| **Data source** | Questionnaires | 9 (69.2) | [25, 27, 29–31, 33–36] |
|  | Interviews | 5 (38.5) | [24, 26–28, 32] |
|  | Focus Groups | 2 (15.4) | [24, 32] |
| **Sampling** | Convenience | 4 (30.8) | [31–33, 36] |
|  | Purposive | 3 (23.1) | [24, 28, 32] |
|  | Snowballing | 1 (7.7) | [32] |
|  | Cluster | 2 (15.4) | [25, 28] |
|  | Unclear | 6 (46.2) | [26, 27, 29, 30, 34, 35] |
| **Population** | Parents/caregivers | 9 (69.3) | [24, 25, 27, 28, 31–35] |
|  | Healthcare professionals | 7 (53.8) | [24, 26, 27, 29, 30, 32, 36] |
|  | Others | 2 (15.4) | [26, 36] |
| **Analysis method** | Descriptive statistics | 8 (61.5) | [25, 27, 29–31, 33–35] |
|  | Thematic analysis | 4 (30.8) | [24, 28, 32, 36] |
|  | Narrative summary | 1 (7.7) | [26] |
| **Time period** | First half of 2020 | 8 (61.5) | [24, 26, 27, 29–31, 35, 36] |
|  | Second half of 2020 | 7 (53.8) | [24, 25, 27, 28, 30, 32, 33] |
|  | Unclear (within 2020) | 1 (7.7) | [34] |

*Characteristics are not mutually exclusive.

## Qualitative studies

Qualitative studies had an almost equal division of reported health-seeking (51.1%) and health-delivery (48.9%) challenges, from a total of 597 calculated responses, including 35 extracted reasons and 92 respondents (**Fig 6**). From these, 11 unique reasons were coded. Respondents were more also more evenly divided between parents (41.3%) and HCPs (58.7), however, most respondents were from either WHO African region (53.3%) or South East Asian region (44.6%). Fear of COVID-19 was a reason reported by all studies [24, 26, 28, 32, 36]; all referred to parents being afraid that they themselves, or their children, might contract COVID-19 during routine immunisation appointments. Most studies also reported inadequate supplies of personal protective equipment (PPE) for HCPs, which was detrimental to the sense of safety amongst both parents and HCPs [26, 28, 32, 36]. Health-delivery challenges included inactive immunisation services [36], challenges with vaccine supplies [36], low levels of staff [36], and lack of guidance regarding the delivery of routine vaccines by HCPs in the context of the COVID-19 pandemic including safety precautions and clear plans for service-provision [32].

Poor availability and access to transport, most likely as a consequence of lockdowns, affected the ability of both parents and HCPs to attend health facilities for immunisations [26, 36] and was, therefore, both a health-delivery and health-seeking challenge, similar to the results seen

**Table 3. Characteristics of each included study.**

| Study reference | Study design | Country | Data collection period (DD/MM/YR) | Data collection methodology | Sampling method | Respondents | Total respondents (sample size) | Method of analysis and synthesis |
|---|---|---|---|---|---|---|---|---|
| Bimpong et al. 2021 [24] | Qualitative | Ghana | 01/03/2020–28/02/2021 | Interview | Purposive | Healthcare professionals | 5 | Thematic |
| | | | | Focus group | Purposive | Caregivers | 4 | |
| Chekhlabi et al. 2021 [29] | Cross-sectional | Morocco | March–June 2020 | Questionnaire | Unclear | Paediatricians | 98 | Descriptive statistics |
| Gupta et al. 2021 [35] | Cross-sectional | India | 10/06/2020–09/07/2020 | Questionnaire | Unclear | Accompanying guardian (parents, grandparents or other) | 249 | Descriptive statistics |
| Hanifi et al. 2021 [26] | Qualitative | Bangladesh | 01/03/2020–31/05/2020 | Interviews | Unclear | Vaccinators | 6 | Narrative summary |
| Khatiwada et al. 2021 [32] | Qualitative | Nepal | 08/2020–12/2020 | Semi-structured interviews | Convenience and snowballing | Service providers | 7 | Thematic |
| | | | | Focus group | Purposive | Service users | 8 | |
| Mahfouz et al. 2021 [34] | Cross-sectional | Egypt | Unclear | Questionnaire | Unclear | Parents of children up to 2 years | 339 | Descriptive statistics |
| Miretu et al. 2021 [25] | Cross-sectional | Ethiopia | 22/07/2020–07/08/2020 | Questionnaire | Multistage cluster | Parents of children aged 15–23 months | 274 | Descriptive statistics |
| Nguyen et al. 2021 [27] | Mixed methods: cross-sectional and qualitative | Bangladesh | 02/2020 and again 09/2020 | Questionnaire and interviews | Unclear | Healthcare professionals | 45 | Descriptive statistics |
| Rizwan et al. 2021 [33] | Cross-sectional | Pakistan | 25/07/2020–07/09/2020 | Questionnaire | Convenience | Parents of children under 2 | 70 | Descriptive statistics |
| Saso et al. 2020 [36] | Qualitative | Multinational (Cameroon, Malawi, Nigeria, South Africa, Sudan, Tanzania, The Gambia, Uganda, Zambia, India, Nepal, Costa Rica, Brazil) | 15/04/2020–30/04/2020 | Questionnaire | Convenience (IMPRINT members) | Healthcare professionals, laboratory-based scientists, public health professionals and other IMPRINT members | 36 | Thematic analysis |
| Shapiro et al. 2022 [31] | Cross-sectional | Multinational (Brazil, China, India, Indonesia, Malaysia, Mexico, Philippines, Thailand, Vietnam) | 14/05/2020–09/06/2020 | Questionnaire | Convenience (YouGov respondents) | Parents | 709 | Descriptive statistics |
| Shet et al. 2021 [30] | Cross-sectional | India | 04/2020-06/2020 and 09/2020 | Questionnaire | Unclear | Healthcare professionals (over 90% paediatricians) | 424 | Descriptive statistics |
| Wale Tegegne et al. 2021 [28] | Qualitative | Ethiopia | 02/09/2020–21/10/2020 | Interviews | Cluster for study areas. Purposive for interviews within areas. | Parents with children aged 10–23 months | 10 | Thematic |

from cross-sectional studies. Additional health-seeking challenges included lockdown policies which discouraged and restricted movement [26, 36], parental concerns over possible vaccine side-effects [24, 36] and parents being misinformed that routine immunisations were not available during the pandemic [24].

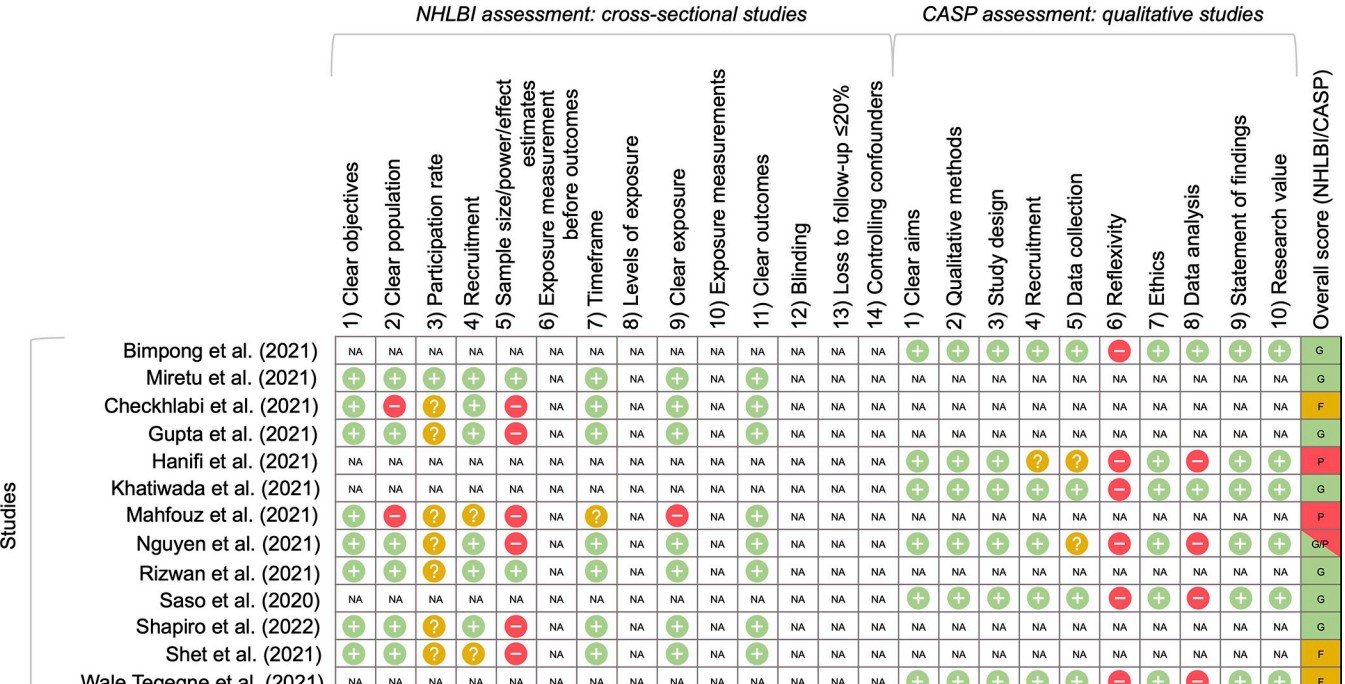

**Fig 2. Results of quality assessment for included studies, following NHLBI and CASP assessments criteria, where (+) denotes that a criterion was met, (-) denotes that it was not met, (?) denotes that it was unclear from the manuscript and supplementary materials, and N/A denotes the criterion was not applicable to the study.** [Overall scores: good (G), fair (F), poor (P)].

One study also explored reasons that did not impact, or encouraged, the continuation of routine child immunisations during the pandemic [32]. This study highlighted that supply of vaccines was not an issue, and that education and encouragement from others was an important factor for both the delivery and the seeking of immunisations, respectively. Specifically, the study suggested HCPs felt a moral obligation to continue providing routine immunisation services and were additionally encouraged by family members to do so. Similarly, some parents continued to seek routine immunisations for their children because they had been educated on the importance of these prior to the COVID-19 pandemic [32]. Another study also found that unavailability of vaccines or poor staffing were not reasons for missed routine immunisations as despite the reallocation of staff towards COVID-19 efforts, routine vaccination services remained staffed and parents did not note any shortages of vaccine supplies [24].

## Discussion

Overall, a total of 17 unique reasons for disruptions were identified, with approximately 80% being healthcare-seeking issues. The most frequently reported reason for not vaccinating children was fear of contracting COVID-19 in healthcare settings, followed by lockdown policies encouraging families to stay home and reduced transport limiting the ability to reach immunisation centres during the pandemic.

Although this study emphasises the importance and influence of factors disrupting healthcare-seeking behaviours, other literature implies healthcare-delivery issues were also significant. Unlike the results from this review, WHO Pulse Survey findings suggest that only 24% of reasons underlying disruptions to health-services stem from decreased demand, the remainder being intentional modifications or unintentional disruptions to services [14]. However, these

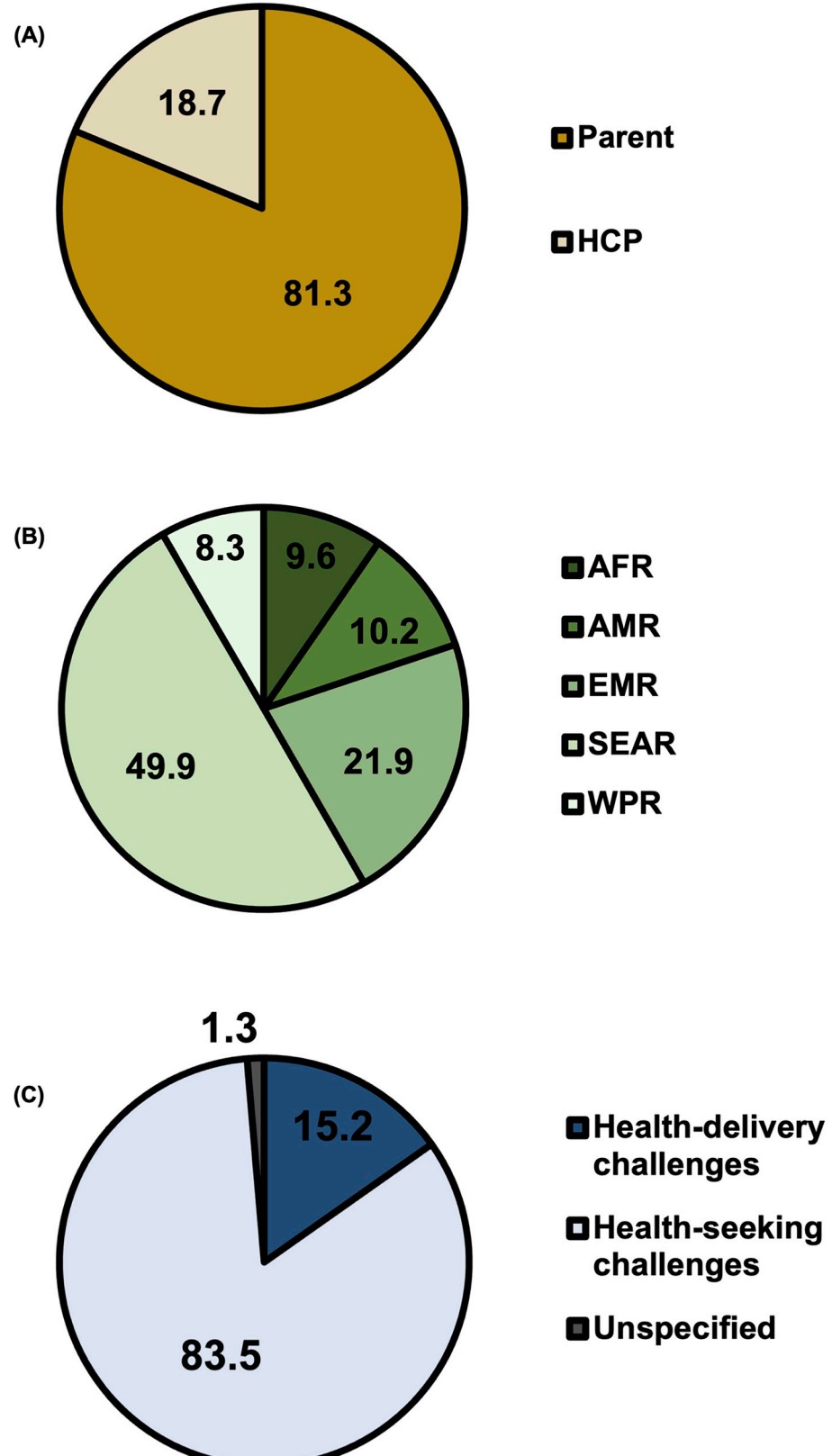

**Fig 3.** Percentage of total respondents in cross-sectional (quantitative) studies by actor (A) and by world region (B); n = 2,853. Percentage of total responses by healthcare-seeking and healthcare-delivery issues (C); n = 3,160.

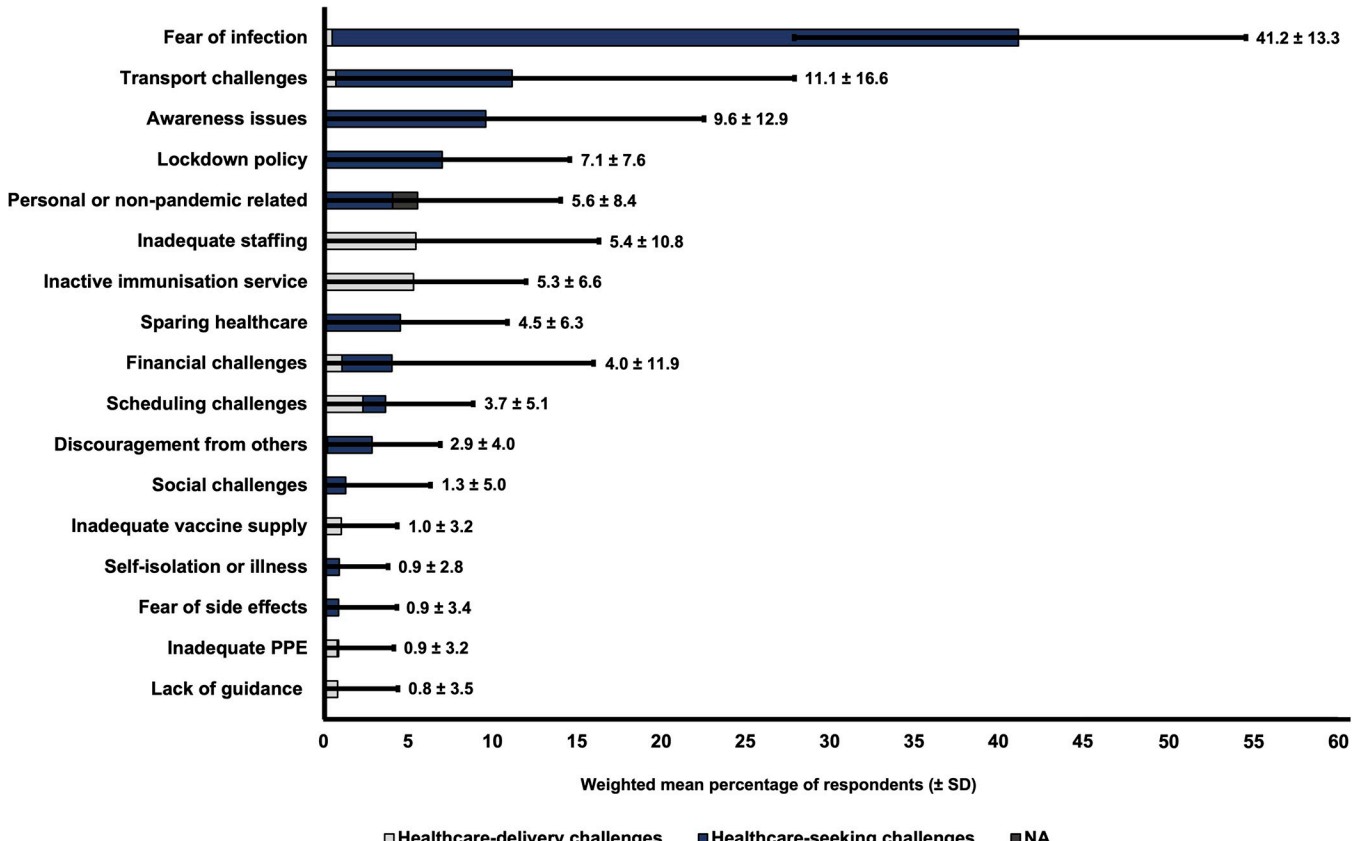

**Fig 4. Weighted mean percentage of total responses (± standard deviation) for each coded reason for disruptions to routine child immunisations, subdivided into healthcare-seeking and healthcare-delivery challenges (n = 3,160).**

proportions apply to health-service disruptions overall, not only immunisation services. The third Pulse Survey also referenced disruptions caused by COVID-19 vaccine rollout [14]; this would not have been captured by this review as studies collected data in 2020, prior to COVID-19 vaccination (**Table 2**). Additionally, in May 2020 UNICEF highlighted falls in vaccine shipments due to travel restrictions and reduced transport, suggesting countries with limited transport options were at particular risk of decreased vaccine supplies [37]. These findings highlight the importance of consulting multiple stakeholders when synthesising evidence for reasons for non-vaccination; approximately 70% of respondents in this study were parents or caregivers, whilst WHO Pulse Surveys contacted Ministry of Health key informants [12, 14] and the IMPRINT respondents were composed of scientists, health workers and other related professionals [36]. It would be expected that parents or caregivers have greater insight into issues affecting health-seeking behaviours, and health workers, scientists and politicians greater insight into healthcare-delivery issues—as seen in the discrepancies above.

Parental fear of COVID-19 appears to be a common cause for missed immunisations universally, including in high income countries (HICs). The multinational study of IMPRINT members included in this review also interviewed members in HICs where fear of COVID-19 was also a common cause for delaying immunisations [36]. Similarly, a cross-sectional study from Saudi Arabia, where parents were surveyed about their children's immunisations, revealed that the most common reason for delaying vaccination was fear of contracting COVID-19 [38].

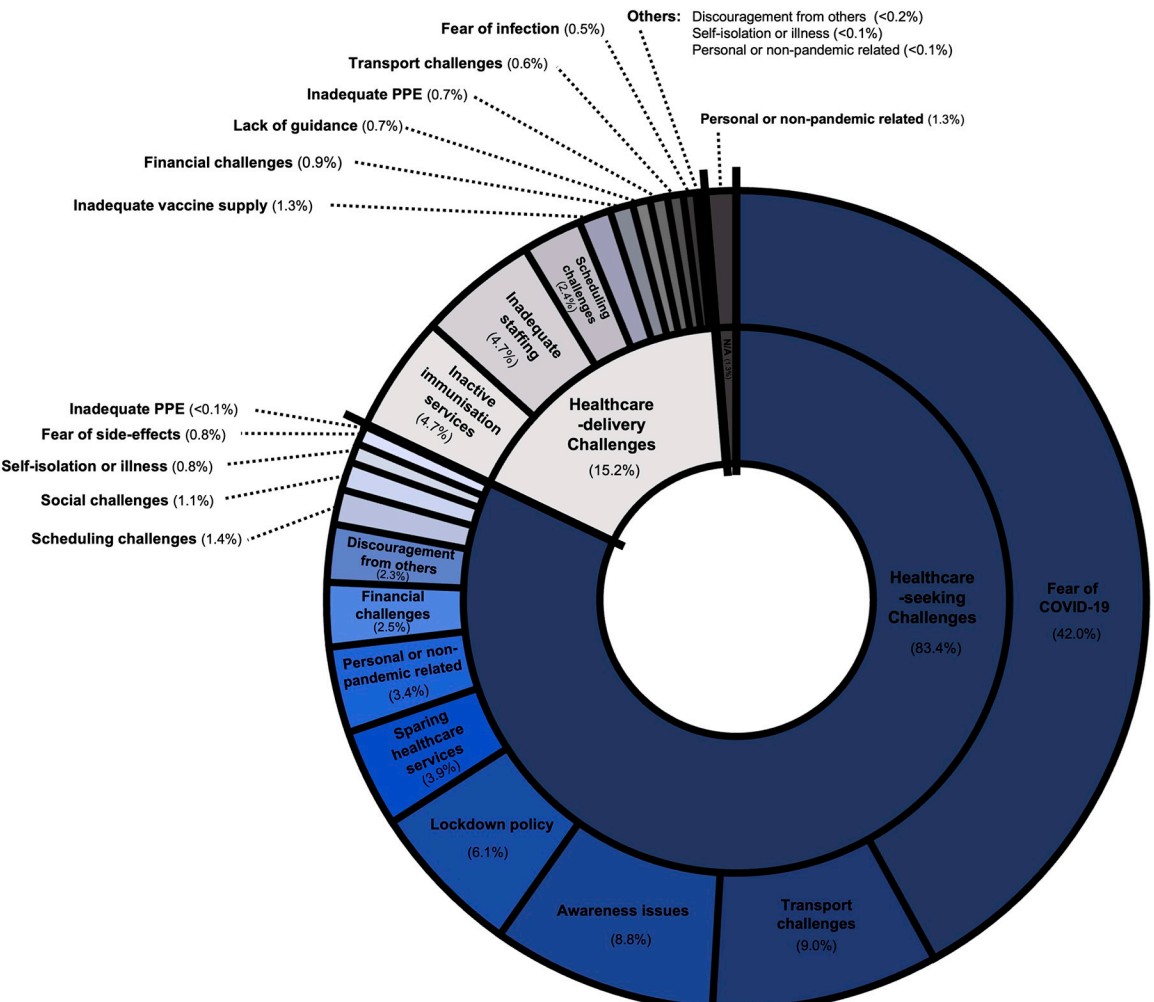

**Fig 5. Coded reasons behind disruptions to routine child immunisations as visual proportions of total responses, subdivided by healthcare-seeking and healthcare-delivery challenges (n = 3,160).** A minority of causes did not apply to either category and were denoted as N/A.

Abbas et al. have highlighted the importance of routine immunisation even if there is a risk of COVID-19 transmission [39]. Their modelling study compared the benefits of routine paediatric immunisations to the risk of SARS-CoV-2 transmission in Africa and found that benefits of immunisations outweighed its risks [39]. For every COVID-19-related death from infection acquired during vaccination, 84 child deaths were prevented. Vaccinated children benefited most, followed by siblings, with elderly family members placed at greatest risk from failure to vaccinate [39]. This evidence emphasises the importance of continuing childhood immunisation programmes throughout the pandemic.

Therefore, as nations develop catch-up programmes to recover lost immunisations, it is essential to combine these efforts with public health campaigns that encourage attendance to routine immunisation services. It is also important to explore methods of delivering routine immunisations that are considered safe and acceptable to families. For instance, ensuring adequate provision of PPE for the public, and enforcing social distancing measures may offer reassurance on the reduced risk of COVID-19 transmission. Alternative strategies for vaccine delivery such as home visits and after-hours programmes may also be more convenient for

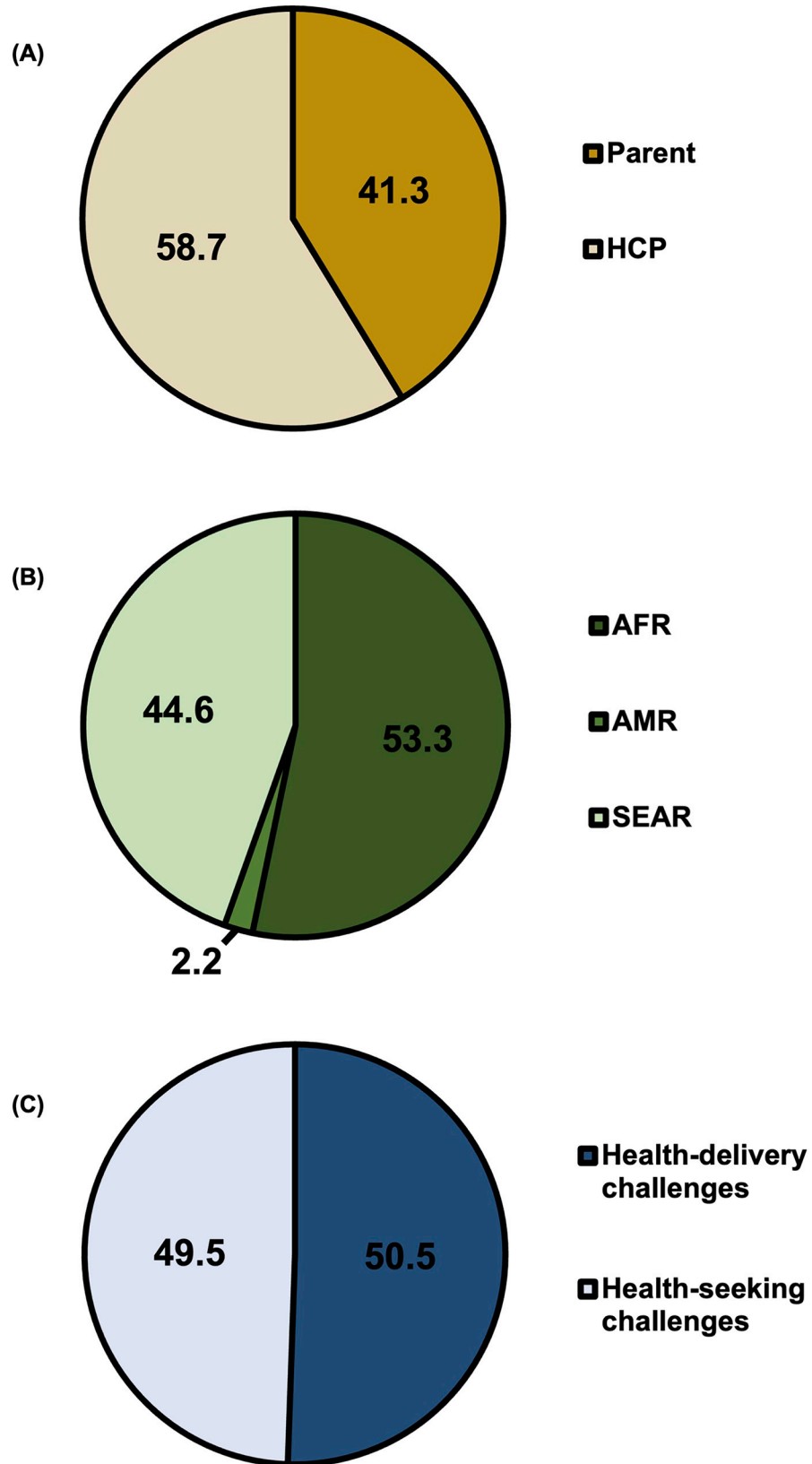

**Fig 6.** Percentage of total respondents in qualitative studies by actor (A) and by world region (B); n = 92. Percentage of total responses by healthcare-seeking and healthcare-delivery issues (C); n = 597.

families and help mitigate the closure of usual vaccine services during lockdown, for instance school programmes [40].

Community engagement is an essential part of addressing reduced health-seeking behaviour [40]. A systematic review analysing the effect of phone reminders to improve routine immunisation uptake in LMICs found overall benefit in these interventions, particularly when combining phone call and text message reminders, although results were highly heterogenous [41]. However, a qualitative study assessing the impact of community engagement programmes in Liberia following the Ebola outbreak in 2014 found that individuals preferred face-to-face engagement and were more likely to follow information from trustworthy sources, such as the radio or community leaders [42]. Individuals also noted the need for involvement of caregivers in educational activities. This campaign led to coverage levels of 99% and over for all vaccines [42]. Another systematic review summarising evidence of interventions to reduce vaccine hesitancy found that multimethod interventions–such as HCP training, social media, and community engagement–and dialogue-based interventions, worked best [43]. In practice, the most effective and feasible methods to reach communities and deliver vaccinations safely will vary between and within countries. Approaches should be adapted to their target population based on local needs, available resources and infection control guidelines.

There are limitations in this review that should be acknowledged. Studies reporting reasons for disruptions relied on self-reporting, which may be subject to recall bias. Studies relying on multiple-choice surveys were also subject to confirmation bias and may not have gathered all reasons for missing or delaying immunisation. Several studies also included participant responses regarding actors other than themselves, such as parents reporting that HCPs were overworked, or HCPs reporting that parents were afraid of COVID-19, limiting their validity. Moreover, despite the review aiming to summarise reasons in LMICs globally, a limited number of countries are represented in this study, and even within each country, only limited samples of the relevant population were included, limiting generalisability. Additionally, the decision to focus on LMICs is in itself very broad, which may challenge the ability to establish specific, effective locally meaningful conclusions; in future, more targeted choices of location or subgroup analyses would be beneficial [44]. There may be local factors that affect disruptions to vaccination, such as differing approaches to COVID-19 mitigation strategies, previous levels of vaccination coverage, accessibility of healthcare and even cultural factors. Therefore, whilst this review may be helpful in providing an overview of key issues, strategies to overcome identified issues must be tailored to local contexts. It is also vital to note that for strategies to be effective and targeted, they should be chosen, designed, and implemented by local teams or authorities.

This review also summarised both quantitative and qualitative findings. Whilst there was substantial overlap in the reasons identified in these two subgroups, quantitative studies had greater proportion of healthcare-seeking issues than qualitative studies. This may be due to the study participants, as there were more HCP respondents in qualitative studies than quantitative, but also likely due to the limitations of assuming that all participants reported each reason (unless otherwise specified) in qualitative studies. When using evidence for policy change, it is vital to include both types of evidence together; quantitative findings may provide a more representative overview of issues, but qualitative findings help provide context and explain these issues more in depth.

Additionally, although the findings of this review focus on pandemic-related drivers of reduced vaccination, vaccine confidence was already a concern prior to the COVID-19 pandemic and hence previously identified, non-pandemic-related factors driving vaccine hesitancy are likely to remain relevant. These may include disbelief in vaccines or severity of vaccine-preventable disease, distrust of health professionals or fear of vaccine side-effects [45, 46].

Discussions for and against the use of COVID-19 vaccines–especially between health professionals and politicians–may have also heightened hesitancy towards other vaccines [47]. One survey-based study found increased hesitancy and heightened perception of routine immunisation risks amongst parents at the Children's Hospital Los Angeles during the pandemic [48]. Furthermore, previously identified barriers to vaccination are likely to also remain relevant, such as living far from immunisation services and associated long duration of travel and associated costs, language and cultural barriers, lack of reminders and poor support from healthcare institutions, amongst several others [45, 49]. This highlights the importance of considering additional factors underlying both vaccine hesitancy and barriers to accessing healthcare when designing efforts for immunisation recovery, and of monitoring the impact of the pandemic on routine paediatric vaccine confidence.

In conclusion, this study shows that most reasons reported for declines in routine child immunisations in LMICs during the COVID-19 pandemic derive from reduced healthcare-seeking behaviours, in particular, fear of COVID-19 and avoidance of healthcare settings. As nations develop efforts to recover missed immunisations, it is essential to invest in measures that ensure continuity of safe vaccine delivery alongside strong public health campaigns advertising availability and safety of routine vaccination to encourage routine immunisation uptake and attendance to health-services, even in the context of a pandemic.

## Supporting information

**S1 Checklist. PRISMA checklists.**
(DOCX)

**S1 Text. Full search strategies.**
(DOCX)

**S2 Text. Changes to protocol.**
(DOCX)

**S1 Table. Definition of each coded reason for disruptions to routine immunisation.**
(DOCX)

**S1 Data.**
(XLSX)

## Acknowledgments

The authors would like to thank Rebecca Jones for the feedback and advice regarding the search strategy.

## Author Contributions

**Conceptualization:** Alexandra M. Cardoso Pinto, Elizabeth Whittaker, James A. Seddon.

**Data curation:** Alexandra M. Cardoso Pinto.

**Formal analysis:** Alexandra M. Cardoso Pinto, Sameed Shariq.

**Investigation:** Alexandra M. Cardoso Pinto, Sameed Shariq, Lasith Ranasinghe.

**Methodology:** Alexandra M. Cardoso Pinto, Shyam Sundar Budhathoki, Helen Skirrow.

**Project administration:** Alexandra M. Cardoso Pinto.

**Resources:** Alexandra M. Cardoso Pinto.

**Software:** Alexandra M. Cardoso Pinto.

**Supervision:** Elizabeth Whittaker, James A. Seddon.

**Validation:** Sameed Shariq, Lasith Ranasinghe, Helen Skirrow.

**Visualization:** Alexandra M. Cardoso Pinto.

**Writing – original draft:** Alexandra M. Cardoso Pinto.

**Writing – review & editing:** Alexandra M. Cardoso Pinto, Sameed Shariq, Lasith Ranasinghe, Shyam Sundar Budhathoki, Helen Skirrow, Elizabeth Whittaker, James A. Seddon.

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
