## [Decision Letter · Decision Letter 0]

2 Nov 2022

PGPH-D-22-01417

Reasons for reductions in routine childhood immunisation uptake during the COVID-19 pandemic in low- and middle-income countries: a systematic review

Dear Dr. Cardoso Pinto,

Thank you for submitting your manuscript to PLOS Global Public Health. After careful consideration, we feel that it has merit but does not fully meet PLOS Global Public Health’s publication criteria as it currently stands. Therefore, we invite you to submit a revised version of the manuscript that addresses the points raised during the review process.

We look forward to receiving your revised manuscript.

Kind regards,

Associate Professor Suman Majumdar

Academic Editor

Journal Requirements:

2. Please insert an Ethics Statement at the beginning of your Methods section, under a subheading 'Ethics Statement'. It must include:

1) The name(s) of the Institutional Review Board(s) or Ethics Committee(s)

2) The approval number(s), or a statement that approval was granted by the named board(s) 

3) (for human participants/donors) - A statement that formal consent was obtained (must state whether verbal/written) OR the reason consent was not obtained (e.g. anonymity). NOTE: If child participants, the statement must declare that formal consent was obtained from the parent/guardian.

4. In the online submission form, you indicated that "All data has been shared as supplementary files. Further clarifications can be made available upon request to corresponding author". All PLOS journals now require all data underlying the findings described in their manuscript to be freely available to other researchers, either 1. In a public repository, 2. Within the manuscript itself, or 3. Uploaded as supplementary information.

Additional Editor Comments (if provided):

Reviewers' comments:

Reviewer's Responses to Questions

**Comments to the Author**

1. Does this manuscript meet PLOS Global Public Health’s publication criteria? Is the manuscript technically sound, and do the data support the conclusions? The manuscript must describe methodologically and ethically rigorous research with conclusions that are appropriately drawn based on the data presented.

Reviewer #1: Yes

Reviewer #2: Yes

2. Has the statistical analysis been performed appropriately and rigorously?

Reviewer #1: N/A

Reviewer #2: Yes

3. Have the authors made all data underlying the findings in their manuscript fully available (please refer to the Data Availability Statement at the start of the manuscript PDF file)?

Reviewer #1: No

Reviewer #2: Yes

4. Is the manuscript presented in an intelligible fashion and written in standard English?

Reviewer #1: Yes

Reviewer #2: Yes

5. Review Comments to the Author

Reviewer #1: Overall, this is a timely review on the impact of COVID-19 on routine immunisation in low and middle income countries globally. Specific comments are provided by each section of the paper below.

Abstract

- Line 41: Some evidence of recovery/bounce back in coverage, including in LMICs, has been noted recently e.g. https://www.gavi.org/news/media-room/immunisation-lower-income-countries-pandemic-leads-decline-coverage-signs-recovery

- Line 50: NHLBI and CASP tools recommended to be spelt out in full at first use

- Line 57: Unclear how the (n=4246) is relevant - not until later in the text that this is reported to be the total number of participants. Suggest removing here or including "with a total of XX participants" after "Thirteen studies..."

- Line 66: 'Reason for disruption' could be made clearer e.g. most common contributors to reduced routine immunisation coverage

- The term 'LMICs' is very broad (and therefore somewhat meaningless), and the included studies hardly consider Central/South America, and do not include any data from the Pacific. While you note this limits generalisability in your discussion, it is important to prevent over-reaching in your framing throughout the article. These articles may also be insightful: https://gh.bmj.com/content/7/6/e009067;
https://gh.bmj.com/content/7/6/e009704

Background

- Line 71: Difficult to assess that children have been spared 'consequences of severe COVID-19'; for instance, what about sick or dying family members, disruptions to schooling etc? While your focus may be on clinical manifestations of COVID-19, this phrasing seems dismissive and I suggest rewording

- Reference 8 does not address declines in routine immunisation due to COVID-19, and reference 9 is too preliminary to provide good evidence to support this. While I believe that COVID-19 impacted on immunisation services, stronger supporting references are needed

- Line 86: Your suggestion that families 'chose' not to vaccinate their children fails to take into account the myriad barriers to accessing routine immunisations and is not supported by global evidence - zero-dose children are overwhelmingly due to access barriers, not hesitancy

Methods

- The link to the previously published paper by the same group (ref 7) should be made more explicit (this could also happen where it is discussed in the background section). At times, the article felt a little lacking substance and splitting this topic in two would help explain - but not justify - why that is the case.

- I am unclear about data availability - there is no attached/supplementary file which includes the calculation of the means for instance. despite stating in the data availability statement that all data have been shared as supplementary files. The completed "pre-defined Excel spreadsheet' (line 142) has also not been shared

- Line 149: is there any justification for why NHLBI and CASP were chosen to assess study quality?

- Line 167: Why do you try to quantify qualitative data? It is a big assumption that, even with a small sample size, every participant reported every issue and is not recommended

Results

- Table 2: The number of studies and the number of references on several lines do not match up (e.g. time period; also extends beyond 2020 in Bimpong et al). Would be good to highlight that the categories are generally not mutually exclusive

- Figure 2: Caption does not adequately explain process. What does +/-/? mean? How were the scores determined? Graphic also unclear/poor quality image

- Line 202: 'failure to immunise' - similar to line 86, the loaded language has strong implications and more considerate terminology regarding the barriers faced should be used

- Fig 4: NA category not clear. Order should match Supp Table A1 (personal barrier category moves position - there should also be some discussion of this later in the paper)

- Line 216: 34 coded causes - does this mean 34 different reasons were identified? How does this fit into the 17 reported for the cross-sectional papers?

- Line 222-3: Unclear if a 'lac of guidance regarding delivery of routine vaccines' is referring to routine immunisation handbooks/guidelines, or Covid-specific information regarding PPE, service provision etc

- Good addition of facilitators to routine immunisation in one study (lines 231-8). Did any other papers include this? Would be interesting to further discussion

Discussion

- Further discussion about the breakdown of specific barriers by respondent status (family or HCP) would be useful - you state in line 166 that barriers are subdivided by reporting stakeholder but this is not clear. For instance, how health-seeking vs health-delivery challenges were determined could do with some more fleshing out - there is some confusion around transport-related barriers

-Line 276-7: how does living far from immunisation services drive vaccine hesitancy? This paragraph would be better later in the discussion. as hesitancy did not appear to be a significant factor identified in your review. Similarly around line 314 the focus on counteracting hesitancy (rather than generally removing barriers to access) seems misplaced

- Line 298-302: Feasibility of all these strategies - providing PPE, home visits, and school programs (especially when schools were shut in many countries for a long period) must be considered in the context of the countries you are discussing - these solutions are often costly and may not be feasible in financially-constrained settings

- Some additional discussion or examples of the types of 'strong public health messaging' you suggest in your concluding paragraph (line 333) would strengthen your discussion

Reviewer #2: This is a thoroughly researched and well written article.

The topic matter is timely and of utmost importance in relation to child health outcomes globally.

Findings should be disseminated widely to promote global efforts to increase routine vaccination in the most vulnerable populations.

Please consider some minor points to review:

Background - sets scene nicely to emphasis drop in immunisation rates, especially in LMICs. Paragraph 2 - consider rephrasing "families chose not to vaccinate" as this implies vaccine hesistancy/anti-vaxxer sentiments and precludes other reasons such as access issues (that you later go onto discuss).

Results

- Overall, there is large difference of proportion between the healthcare seeking challenges reasons (81.9%) and the healthcare delivery challenges (15.8%). It might make more impact on this difference if you present results this way. This is demonstrated /visualised well in figure 5 but not discussed that way in your written results section.

- your qualitative studies show relative equal proportion between health seeking and health delivery reasons. But I can't see what the proportions are in the quantitative studies? I assume more weighted towards health seeking challenges to have overall results tip that way.

Discussion

- Has the difference in ways data is collected between quantitative and qualitative studies affected the results? If findings are quite disparate, which type of studies should we rely more on for policy/program development?

- The results included WHO regions and countries where the studies were from. However, there is no analysis of this in the discussion. Are there particular features/ aspects from these countries (eg. cultural beliefs, local policy on COVID mitigation methods), that may have affected the results? As these reasons would help targeted interventions for those countries/regions.

References

Ensure references follow appropriate referencing style (Vancouver). There shouldn't be pdfs attached to end of reference; websites/online articles should have date cited and "available from" prior to URL

6. PLOS authors have the option to publish the peer review history of their article (what does this mean?). If published, this will include your full peer review and any attached files.

**Do you want your identity to be public for this peer review?** For information about this choice, including consent withdrawal, please see our Privacy Policy.

Reviewer #1: No

Reviewer #2: No

---

## [Decision Letter · Decision Letter 1]

3 Jan 2023

Reasons for reductions in routine childhood immunisation uptake during the COVID-19 pandemic in low- and middle-income countries: a systematic review

PGPH-D-22-01417R1

Dear Miss Cardoso Pinto,

We are pleased to inform you that your manuscript 'Reasons for reductions in routine childhood immunisation uptake during the COVID-19 pandemic in low- and middle-income countries: a systematic review' has been provisionally accepted for publication in PLOS Global Public Health.

Best regards,

Associate Professor Suman Majumdar

Academic Editor

Reviewer Comments (if any, and for reference):

Reviewer's Responses to Questions

**Comments to the Author**

1. If the authors have adequately addressed your comments raised in a previous round of review and you feel that this manuscript is now acceptable for publication, you may indicate that here to bypass the “Comments to the Author” section, enter your conflict of interest statement in the “Confidential to Editor” section, and submit your "Accept" recommendation.

Reviewer #1: All comments have been addressed

Reviewer #2: All comments have been addressed

2. Does this manuscript meet PLOS Global Public Health’s publication criteria? Is the manuscript technically sound, and do the data support the conclusions? The manuscript must describe methodologically and ethically rigorous research with conclusions that are appropriately drawn based on the data presented.

Reviewer #1: Yes

Reviewer #2: Yes

3. Has the statistical analysis been performed appropriately and rigorously?

Reviewer #1: Yes

Reviewer #2: Yes

4. Have the authors made all data underlying the findings in their manuscript fully available (please refer to the Data Availability Statement at the start of the manuscript PDF file)?

Reviewer #1: Yes

Reviewer #2: Yes

5. Is the manuscript presented in an intelligible fashion and written in standard English?

Reviewer #1: Yes

Reviewer #2: Yes

6. Review Comments to the Author

Reviewer #1: The authors have meaningfully engaged with comments from the previous review and the revised manuscript clearly highlights how these changes were taken into consideration. Thank you for making your data available as well. The numbering of the supplementary sections may need to be updated to match the new formatting. This is a timely and interesting piece of work, and I have no further comments.

Reviewer #2: Thank you for revising the article and addressing all comments in a succinct manner. The article now provides improved clarity of results and stronger messages in the discussion.

7. PLOS authors have the option to publish the peer review history of their article (what does this mean?). If published, this will include your full peer review and any attached files.

**Do you want your identity to be public for this peer review?** For information about this choice, including consent withdrawal, please see our Privacy Policy.

Reviewer #1: No

Reviewer #2: No
